# Zero-Shot Self-Supervised Joint Temporal Image and Sensitivity Map Reconstruction via Linear Latent Space

**Molin Zhang**[1]                 MOLIN@MIT.EDU
[1] *Department of Electrical Engineering and Computer Science,*
*Massachusetts Institute of Technology, Cambridge, MA, USA*

**Junshen Xu**[1]                JUNSHEN@MIT.EDU
**Yamin Arefeen**[1]            YAREFEEN@MIT.EDU

**Elfar Adalsteinsson**[1,2,3]        ELFAR@MIT.EDU
[2] *Harvard-MIT Health Sciences and Technology, Massachusetts Institute of Technology,*
*Cambridge, MA, USA*
[3] *Institute for Medical Engineering and Science, Massachusetts Institute of Technology,*
*Cambridge, MA, USA*

**Editors:** Accepted for publication at MIDL 2023

## Abstract

Fast spin-echo (FSE) pulse sequences for Magnetic Resonance Imaging (MRI) offer important imaging contrast in clinically feasible scan times. T2-shuffling is widely used to resolve temporal signal dynamics in FSE acquisitions by exploiting temporal correlations via linear latent space and a predefined regularizer. However, predefined regularizers fail to exploit the incoherence especially for 2D acquisitions.Recent self-supervised learning methods achieve high-fidelity reconstructions by learning a regularizer from undersampled data without a standard supervised training data set. In this work, we propose a novel approach that utilizes a self supervised learning framework to learn a regularizer constrained on a linear latent space which improves time-resolved FSE images reconstruction quality. Additionally, in regimes without groundtruth sensitivity maps, we propose joint estimation of coil-sensitivity maps using an iterative reconstruction technique. Our technique functions is in a zero-shot fashion, as it only utilizes data from a single scan of highly undersampled time series images. We perform experiments on simulated and retrospective in-vivo data to evaluate the performance of the proposed zero-shot learning method for temporal FSE reconstruction. The results demonstrate the success of our proposed method where NMSE and SSIM are significantly increased and the artifacts are reduced.

**Keywords:** Temporal MR image reconstruction, Linear latent space, Zero-shot supervised learning, Joint sensitivity map estimation.

## 1. Introduction

Fast spin-echo (FSE) Magnetic Resonance Imaging (MRI) sequences provide a variety of image contrasts at different echo times in clinically reasonable scan times (Hennig et al., 1986; Mugler III, 2014). While standard FSE reconstructs a single image, recent techniques aim to resolve a time-series of images during each echo of a FSE acquisition.

T2-shuffling has demonstrated success in resolving temporal images from volumetric FSE acquisitions (Tamir et al., 2017; Bao et al., 2017; Iyer et al., 2021). T2 shuffling randomly shuffles the phase encode view ordering, exploits temporal correlations with a

low-dimensional subspace model, and utilizes a predefined regularizer (Lustig et al., 2008; Zhao et al., 2015). However, predefined regularizers fail to exploit the limited achievable incoherence in 2D-acquisition which precludes widespread use of T2-shuffling in 2D FSE.

Recently, deep learning (DL) has gained interest for high-quality accelerated MRI by combining a learned regularization and the data consistency term from deep neural networks in an unrolled manner (MoDL) (Aggarwal et al., 2018). However, MoDL requires a large scale training dataset of under-sampled data with associated ground truth, which may be unrealistic to acquire in clinical practice. To train models without ground truth, Yaman et al. (Yaman et al., 2020b,a, 2021) proposed a self-supervised learning reconstruction method (SSDU), which trains models in a self-supervised fashion by partitioning under-sampled kspace data into two disjoint sets, $\Theta$ and $\Lambda$, and training a network as a regularizer in the traditional optimizations with the information from $\Theta$ to predict the unseen data, $\Lambda$.

Moreover, current reconstruction methods require groundtruth coil sensitivity maps, usually estimated from fully sampled calibration regions (ESPIRIT) (Uecker et al., 2014). To further shorten total acquisition times or to handle scenarios with no calibration data available, joint reconstruction approaches are proposed to estimate coil sensitivity maps and in the unrolled iterative reconstruction of MRI (Jun et al., 2021; Arvinte et al., 2021).

We propose a novel zero-shot self-supervised learning technique that jointly reconstructs coil sensitivity maps and a time-series of 2D images on a linear latent space from a single T2-shuffling acquisition. Our contributions are the following: 1) We trained our model in a zero-shot, self-supervised fashion from just a single series of highly under-sampled 2D MR images with T2-shuffling acquisitions. 2) We utilized the temporal linear latent space to improve the condition of the reconstruction problem along with random masking to mitigate over-fitting. 3) If lack of calibration data for the estimate of the groundtruth sensitivity maps, our zero-shot self-supervised framework jointly estimates coil sensitivity maps and the time-series of images.

We evaluate the performance of our model in two scenarios: First, only reconstruct the highly undersampled temporal images where the groundtruth sensitivity maps are provided; Second, jointly estimate the sensitivity maps and reconstruct the temporal images from the data where the groundtruth sensitivity maps are lack.We demonstrate the success of our model on both simulated data and in-vivo scanning data with different sampling patterns . The overall image qualities are significantly improved.

## 2. Methods

### 2.1. Temporal linear latent model

For a 2D time series of MR images with FSE acquisition, temporal signal evolution of each voxel (tissue) with differing (T2, T1) parameters has strong correlation. A linear low-rank latent approximation of the signal correlation will help reduce the degrees of freedom in the time-resolved reconstruction problem (Tamir et al., 2017; Liang, 2007; Haldar and Liang, 2011). The details are shown below:

FSE signal evolution can be simulated with the Extended Phase Graph (EPG) (Hennig, 1988; Ben-Eliezer et al., 2015) algorithm for a certain (T2, T1) and flip angle. A dictionary containing an ensemble of signal evolution $\mathbf{Q} \in \mathbb{C}^{T \times N}$ from a range of (T2, T1) values and the flip angle of interest is generated, where $T$ is the number of echos and $N$ is the number

of different tissue parameters. The linear latent space $\mathbf{\Phi}$ is derived from $\mathbf{Q}$ to exploit the general temporal correlation between different signals.

As depicted in Fig. 1(a), applying singular value decomposition (SVD) to $\mathbf{Q}$ and taking the singular vectors of the top $K$ largest singular values produces a linear latent space model $\mathbf{\Phi} \in \mathbb{C}^{T \times K}$, which more compactly represents FSE signal evolution. By constraining the time-series of images to the linear latent space, $\mathbf{X_T} = \mathbf{\Phi}\boldsymbol{\alpha}$, where $\mathbf{X_T} \in \mathbb{C}^{T \times W \times H}$ are the pixels to be reconstructed in the temporal images and $\boldsymbol{\alpha} \in \mathbb{C}^{K \times W \times H}$ are coefficients, previous works (Tamir et al., 2017) solve the reconstruction problems on the coefficients $\boldsymbol{\alpha}$:

$$\min_{\boldsymbol{\alpha}} \|\mathbf{Y} - \mathbf{MFS\Phi\alpha}\|_2^2 + \mu R(\boldsymbol{\alpha}) \tag{1}$$

where $\mathbf{Y}$ is the acquired data, $\mathbf{M}$ is the sampling pattern, $\mathbf{F}$ is the Fourier transform, $\mathbf{S}$ is the sensitivity map, $\mu$ is a hyperparameter, and $R$ is a predefined regularizer on the temporal coefficients $\boldsymbol{\alpha}$.

### 2.2. Zero-shot self-supervised image reconstruction on linear latent space

For 2D FSE acquisition with high under-sampling rate, predefined regularizers fail to exploit the limited achievable incoherence. Recent self-supervised frameworks learn the regularizer directly from the acquired, under-sampled data (Eq. 2), without prior knowledge or ground truth, and combine the learned regularization with a data consistency term (Eq. 3) using an unrolled iterative reconstruction algorithm (Aggarwal et al., 2018).

$$\mathbf{Z}^{(i-1)} = \mathcal{D}_{\mathbf{I}}\left(\mathbf{X}^{(i-1)}\right) \tag{2}$$

$$\mathbf{X}^{(i)} = \arg\min_{\mathbf{X}} \|\mathbf{Y} - \mathbf{MFSX}\|_2^2 + \mu \left\|\mathbf{X} - \mathbf{Z}^{(i-1)}\right\|_2^2 \tag{3}$$

$$= \left((\mathbf{MFS})^H (\mathbf{MFS}) + \mu\mathbf{I}\right)^{-1} \left((\mathbf{MFS})^H \mathbf{Y} + \mu\mathbf{Z}^{(i-1)}\right) \tag{4}$$

where $\mathcal{D}_{\mathbf{I}}$ is a trainable network as a regularizer generator where the reconstructed image $\mathbf{X}^{(i-1)}$ is the input and $\mathbf{Z}^{(i-1)}$ is the output regularizer, at the $i$-th iteration. We force reconstructed image $\mathbf{X}^{(i)}$ at next step $i$ as close to $\mathbf{Z}^{(i-1)}$ as possible, controlled by the hyperparameter $\mu$. The closed-form serves as a solution to the data consistency step with learnt regularizer (Eq. 3) requires an infeasible matrix inversion. Instead, we use the Conjugate Gradients (CG) method to solve Eq. 4.

To learn the regularization function without access to fully-sampled ground truth, the self-supervised framework divides the acquired kspace $\Omega$ into two disjoint sub-kspaces $\Omega = \Theta \cup \Lambda$ (Yaman et al., 2020b,a, 2021). As shown in Fig. 1(b), the sub-kspace $\mathbf{Y}_\Theta$ is used for the data consistency term, i.e., $\mathbf{Y}$ and $\mathbf{M}$ in Eq. 3 are replaced by $\mathbf{Y}_\Theta$ and $\mathbf{M}_\Theta$ respectively. The other sub-kspace $\mathbf{Y}_\Lambda$ is used for computing the the loss $\mathcal{L}(\mathbf{u}, \mathbf{v}) = \frac{\|\mathbf{u}-\mathbf{v}\|_2}{\|\mathbf{u}\|_2} + \frac{\|\mathbf{u}-\mathbf{v}\|_1}{\|\mathbf{u}\|_1}$ between $\mathbf{Y}_\Lambda$ and the predicted kspace $\mathbf{M}_\Lambda \mathbf{FS\hat{X}}$ in the same $\Lambda$ region for the network $\mathcal{D}_{\mathbf{I}}$ training, where $\hat{\mathbf{X}}$ is the output of the final unrolled block with the information from $\mathbf{Y}_\Theta$.

However, current self-supervised framework shows success only on single image reconstruction but fails at highly under-sampled (4 or 8 times lower than conventional single image) FSE images. In this work, we applied the self-supervised framework of the learnt

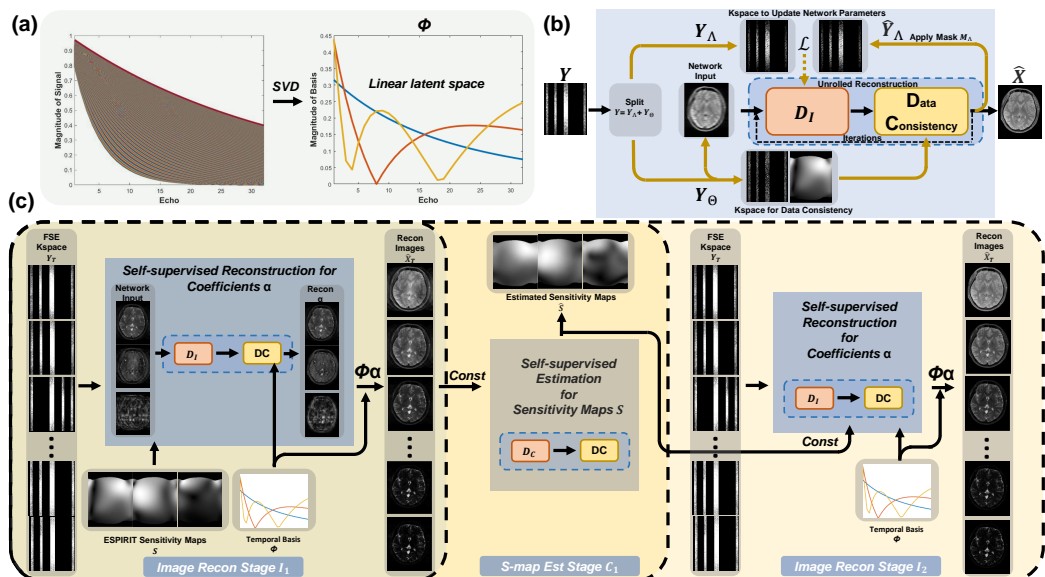

Figure 1: An overview of our proposed reconstruction framework. **(a)**: The linear latent space is generated by performing SVD on a signal evolution dictionary matrix and selecting the eigenvectors with the top $K$ largest eigenvalues. **(b)**: The illustration of the self-supervised recon process. **(c)**: The illustration of our proposed framework. There are two image recon stages and one s-map estimation stage. The zero-shot self-supervised process in the image stage $I_i$ is applied on the linear latent space $\boldsymbol{\Phi}$ by exploiting the temporal correlations to improve the condition of the problem. The coefficients $\boldsymbol{\alpha}$ of the linear latent space are reconstructed instead of images. The temporal images $\hat{\mathbf{X}}_{\mathbf{T}}$ could be recovered by $\boldsymbol{\Phi}\boldsymbol{\alpha}$ and are used for the s-map estimation stage $C_1$ to estimate s-maps $\hat{S}$. The estimated s-maps $\hat{S}$ are used for the following image reconstruction stage $I_2$ with the linear latent space as well. Note that we stop the gradient flow and freeze the trainable networks $D_I$ and $D_C$ between stages by setting $\hat{\mathbf{X}}_{\mathbf{T}}$ and $\hat{\mathbf{S}}$ as constant.

regularizer to the linear latent space $\boldsymbol{\Phi}$ to optimize the coefficients $\boldsymbol{\alpha}$ in a zero-shot manner for the first time. Our framework enables reconstruction of time-resolved MRI data with significantly reduced degrees of freedom.

Our reconstruction framework is formulated as below,

$$\mathbf{Z}_{\mathbf{I}}^{(i-1)} = \mathcal{D}_{\mathbf{I}}\left(\boldsymbol{\alpha}^{(i-1)}\right) \tag{5}$$

$$\boldsymbol{\alpha}^{(i)} = \arg\min_{\boldsymbol{\alpha}} \|\mathbf{Y}_{T\Theta} - \mathbf{M}_{\Theta}\mathbf{F}\mathbf{S}\boldsymbol{\Phi}\boldsymbol{\alpha}\|_2^2 + \mu_I \left\|\boldsymbol{\alpha} - \mathbf{Z}_{\mathbf{I}}^{(i-1)}\right\|_2^2 \tag{6}$$

where $\mathbf{Y}_{T\Theta} \in \mathbb{C}^{T \times W \times H \times C}$ is the temporal kspace for the data consistency. $C$ is the number of sensitivity maps. We use the same loss $\mathcal{L}(\mathbf{u}, \mathbf{v})$ to update the parameters in the network

as described above. The temporal FSE images is reconstructed from the linear latent space using $\mathbf{X_T} = \mathbf{\Phi}\hat{\boldsymbol{\alpha}}$, where $\hat{\boldsymbol{\alpha}}$ is the output of the final unrolled block.

In addition, we apply our technique in a zero-shot fashion where we only have access to the under-sampled data to reconstruct. In order to mitigate overfitting, we randomly generate a new pair of disjoint masks for each training epoch (Quan et al., 2020).

The proposed framework requires much higher GPU memory for back-propagation related to the data consistency term due to the additional echo dimension. We use CG to approximate the gradient flow of the data consistency term, where

$$\left(\nabla_{\mathbf{Z_I}^{(i-1)}}\mathcal{L}\right) = \left((\mathbf{M}_\Theta\mathbf{FS}\mathbf{\Phi})^H\,(\mathbf{M}_\Theta\mathbf{FS}\mathbf{\Phi}) + \mu_I\mathcal{I}\right)^{-1}\left(\nabla_{\boldsymbol{\alpha}^{(i)}}\mathcal{L}\right) \tag{7}$$

### 2.3. Self-supervised sensitivity map estimation:

Standard algorithms, like ESPIRIT (Uecker et al., 2014), estimate coil sensitivity maps $\mathbf{S}$, from fully sampled auto-calibration scans (ACS). To shorten total scan time or handle situations without groundtruth sensitivity maps, we propose another unrolled self-supervised method to reconstruct sensitivity maps directly from the acquired FSE data.

$$\mathbf{Z_C}^{(i-1)} = \mathcal{D}_\mathbf{C}\left(\mathbf{S}^{(i-1)}\right) \tag{8}$$

$$\mathbf{S}^{(i)} = \arg\min_{\mathbf{S}} \|\mathbf{Y}_{T\Theta} - \mathbf{M}_\Theta\mathbf{FX_TS}\|_2^2 + \mu_C\left\|\mathbf{S} - \mathbf{Z_C}^{(i-1)}\right\|_2^2 + \lambda_C\|\mathbf{DS}\|_2^2 \tag{9}$$

where $\mathbf{X_T}$ are the temporal images and $\mathbf{D}$ is the spatial gradient operator which imposes smoothness constraints on the reconstructed sensitivity map. $\mathcal{D}_\mathbf{C}$ is another trainable network to generate the learnt regularizer for the sensitivity maps. We switch the order of $\mathbf{X_T}$ and $\mathbf{S}$ because of the element-wise product. Eq. 9 can still be solved by CG algorithm. Same loss $\mathcal{L}(\mathbf{u}, \mathbf{v})$ is used to update the parameters in $\mathcal{D}_\mathbf{C}$.

We found simultaneously reconstructing the images and sensitivity maps to be unstable and sensitive to initialization. Instead, as presented in Fig. 1 (c), the image reconstruction stage $I_i$ and the sensitivity map estimation stage $C_i$ are trained stage-by-stage. In this work, we use two image stages and one s-map stage. Each stage is trained for $N$ steps. Each image/sensitivity stage uses the results from the previous sensitivity/image stage while the first image stage applies sensitivity maps estimated from ESPIRIT, which utilizes the central $k$ lines in the acquired FSE data. To avoid overfitting, the sampling mask for the blocks are generated randomly and independently based on a Gaussian distribution. The ratio of the cardinality of $|\Lambda|$ and $|\Theta|$ is 2/3. $\mathcal{D}_\mathbf{C}$ is frozen and stops update when $\mathcal{D}_\mathbf{I}$ is updated in the image stage, vice versa.

## 3. Experiments and Results

### 3.1. Dataset

We evaluate our proposed reconstruction method on both simulated and retrospective in-vivo 2D FSE data. The FSE protocols are different between the simulated and the in-vivo data to show the generalization of our proposed method.

For the simulated data, the EPG algorithm simulates a FSE sequence with (8 coils, T=80 echoes, echo spacing of 5.56 ms, matrix size of $190 \times 256$ and $160°$ refocusing train) based on numerical brain T2 and proton density maps (Collins et al., 1998).

For the retrospective in-vivo data, we acquired a fully-sampled spatial-temporal multi-echo dataset on human brain (FOV=180mm$\times$240mm, matrix size of $208 \times 256$, slice thickness=3mm, echo spacing=11.5ms, T=32 echoes, 12 coils) (Zhao et al., 2015).

### 3.2. Experimental setup

The EPG algorithm simulates a dictionary of FSE signal evolution with a T2 range of 5 - 400 ms, T1 = 1000 ms, and refocusing angle = 160, 180 for the simulated and in-vivo datasets respectively. Then, the SVD of the dictionary generates a basis for the linear latent space with $K = 3$.

To generate T2-shuffling data, we under-sampled each echo in the simulated dataset by $R = 24$ (8 phase encoding lines/echo) and in the in-vivo dataset by $R = 16$ (13 phase encoding lines/echo). If lack of groundtruth sensitivity maps, we sampled $\{2, 4, 6\}$ (simulated data) and $\{2, 4, 6, 8, 10\}$ (in-vivo data) phase encoding (ky) lines at the center of the kspace for the ESPIRIT to estimate rough sensitivity maps while ensuring that the acceleration rate stayed at either R=24 or 16. To model a T2-shuffling acquisition, the non-calibration k-space lines were randomly distributed throughout the echo train.

For model training, we train the image stage $I_1$ for 100 steps, $I_2$ for 100 steps, and the s-map stage $C_1$ for 100 steps with an Adam (Kingma and Ba, 2014) optimizer. Both stages use resnet as the backbone for the regularizer generator. For the image stage $I_1$, the learning rate is 5e-4 and 1e-4 for the simulated and retrospective experiments respectively (both decay to 5e-5 after 40 steps). For all experiments, we use 5e-5 for the image stage $I_2$ and 5e-5 for the s-map stage $C_1$, regularization parameters $\mu_I = 0.05$, $\mu_C = 0.02$ and $\lambda_C = 2.0$, and 10 unrolled block.

We compare four methods: 1) 2D T2 shuffling reconstruction (Shuffling), 2) SSDU with individual images (SSDU), 3) Zero-shot self-supervised image reconstruction with linear latent space (Ours-sub) and 4) our proposed joint image and sensitivity map reconstruction with the linear latent space (Ours-joint). There are two settings: i) groundtruth sensitivity maps are provided where only Ours-sub is evaluated and ii) lack of groundtruth s-maps where Ours-sub and Ours-joint are both evaluated. We repeat each experiment 5 times to report the mean and variance of quantitative metrics. For the Shuffling method, we choose l1-wavelet as the regularization (Iyer et al., 2021). All experiments are performed on an NVIDIA TESLA V100 for 40 mins.

### 3.3. Results

We evaluate performance by measuring the Normalized Mean Square Error (NMSE) and the Structural Similarity (SSIM) (Wang et al., 2004) of the reconstructed magnitude images and the NMSE of the estimated T2 maps, since FSE sequences yield echoes with constant phase.

**I. Simulated data** Table 1 shows quantitative performance of the different methods with just two center ky lines for the experiments without groundtruth s-maps, which closely represents a standard T2-shuffling acquisition. With groundtruth coil-sensitivity maps, our

Table 1: Mean NMSE (%) and SSIM of the reconstructed temporal images and estimated T2 maps with 2 center ky lines. NMSE-I and SSIM-I stand for the NMSE and SSIM of the images, NMSE-T2 stands for the NMSE of the T2 maps. ⋆ represents the presence of groundtruth sensitivity maps. Sf stands for the Shuffling method.

| Data | Metric | Sf⋆ | SSDU⋆ | Sub⋆ | Sf | Sub | Joint |
|---|---|---|---|---|---|---|---|
| Simulated | NMSE-I (%) | 9.16 | 42.3 | **7.23** | 19.3 | 18.8 | **11.2** |
| | SSIM-I | 0.945 | 0.693 | **0.964** | 0.895 | 0.910 | **0.942** |
| | NMSE-T2 (%) | 24.7 | 35.5 | **19.1** | 29.7 | 26.8 | **23.5** |
| In-vivo | NMSE-I (%) | 16.2 | 35.1 | **11.9** | 19.5 | 17.2 | **16.2** |
| | SSIM-I | 0.942 | 0.840 | **0.950** | 0.922 | 0.924 | **0.928** |
| | NMSE-T2 (%) | 18.4 | 28.4 | **16.5** | 25.5 | 27.1 | **21.3** |

proposed technique improves image NMSE and SSIM, and T2 NMSE in comparison to the baseline techniques. Without gt sensitivity maps, the proposed joint estimation of images and sensitivity maps outperforms shuffling and the non-joint proposed algorithm.

Fig. 2(a) displays example reconstructed temporal images and estimated T2-maps. Again, with groundtruth sensitivity maps, the proposed technique yields images and T2-maps with lower NMSE, improved SSIM, and qualitatively less artifacts. Without gy sensitivity maps, the ESPIRIT algorithm produces maps that lead to signal bias. Our proposed joint reconstruction method estimates better sensitivity maps, leading to higher fidelity image reconstructions. Fig. 2(b) plots average SSIM and NMSE across all echoes for sampling patterns with different numbers of central k-space lines. Our proposed technique readily integrates with different sampling patterns and out-performs the baselines in all cases.

**II. In-vivo data**

Table 1 and Fig. 3(a) show average performance metrics and reconstructed examples from the sampling pattern with 2 center ky lines in the in-vivo data. Even with imaging parameters that vary from the simulation experiments, the proposed methods outperform the compared methods quantitatively. Qualitatively, shuffling suffers from a variety of artifacts while the proposed methods more closely resembles ground truth. Additionally, without groundtruth sensitivity maps, the proposed joint reconstruction method produces images with improved NMSE and SSIM and less imaging artifacts.

As seen in Fig. 2(b) and 3(b), the proposed method outperforms the baselines with sampling patterns using various numbers of center ky lines. Performance does not necessarily improve with more center ky lines as each sampling pattern uses the same overall acceleration rate, and the trade-off between center and external k-space lines for performance differs across various dataset.

## 4. Conclusion

In this work we proposed a novel zero-shot self-supervised method for time-resolved MRI reconstruction from 2D FSE acquisitions. Our proposed approach constrains a self-supervised

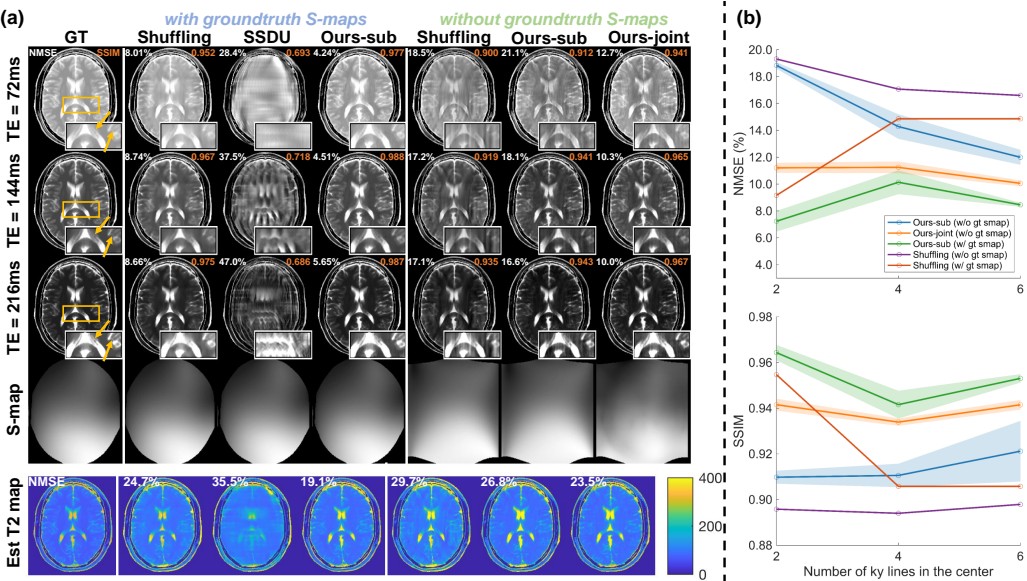

Figure 2: **(a)**: Examples of the reconstructed temporal images and estimated T2 maps on the simulated data with 2 center phase encoding lines. The NMSE and SSIM are calculated based on the magnitude of the images. To have a clear view of the details, we modify the lightness of the images. As SSDU fails to reconstruct the image, we only ran the experiment of SSDU once. **(b)**: The mean NMSE and SSIM of the reconstructed temporal images with different numbers of center phase encoding lines on the simulated data. The solid lines are the average metrics and the filled regions represent the value of the standard deviation (std) plus-minus of the metrics.

reconstruction framework on a linear latent space to simultaneously learn a regularizer from the highly under-sampled data itself and exploit temporal correlations to significantly reduce degrees of freedom in the reconstruction. Moreover, a self-supervised sensitivity estimation stage is proposed which only utilizes the acquired data to further shorten the total scanning time.

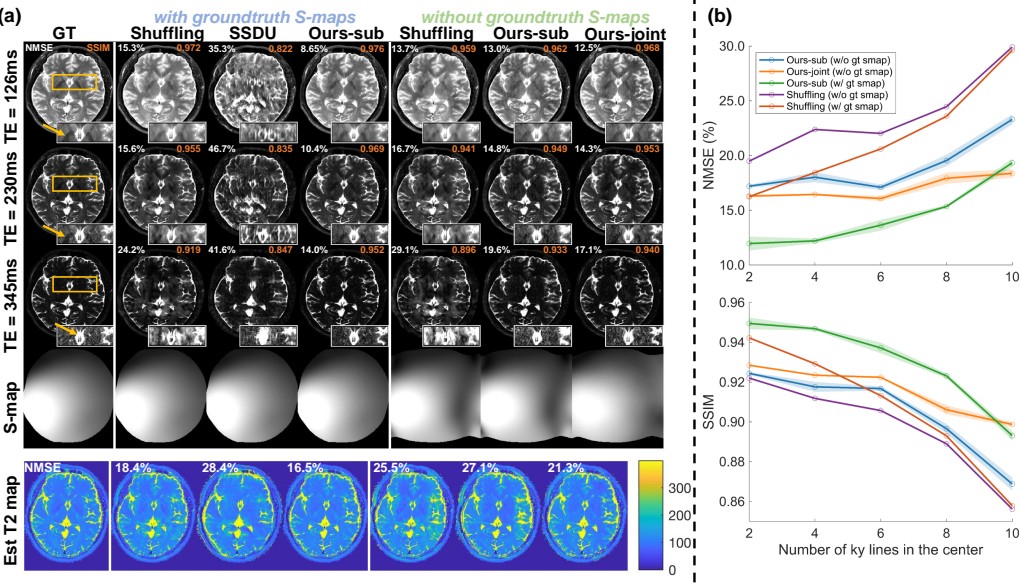

Figure 3: **(a)**: Examples of the reconstructed temporal images and estimated T2 maps on the in-vivo data with 2 center phase encoding lines. We also modify the lightness of the images to have a better view. SSDU is only run once. **(b)**: The mean NMSE and SSIM of the reconstructed temporal images with different numbers of center phase encoding lines on the in-vivo data.

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

# Appendix A. The effect of the size of the linear latent space

In this appendix section, the effect of the size of the linear latent space $\mathbf{\Phi} \in \mathbb{C}^{T \times K}$ is studied where the number of the singular vector $K$ is changed. In our work, we choose $K = 3$. As shown in fig. 4, the top three singular values account for 99.6% of the sum of all the singular values which recover the original dictionary with a adequately small normalized error of 5%, while $K = 4$ singular vectors recover with 0.6% normalized error.

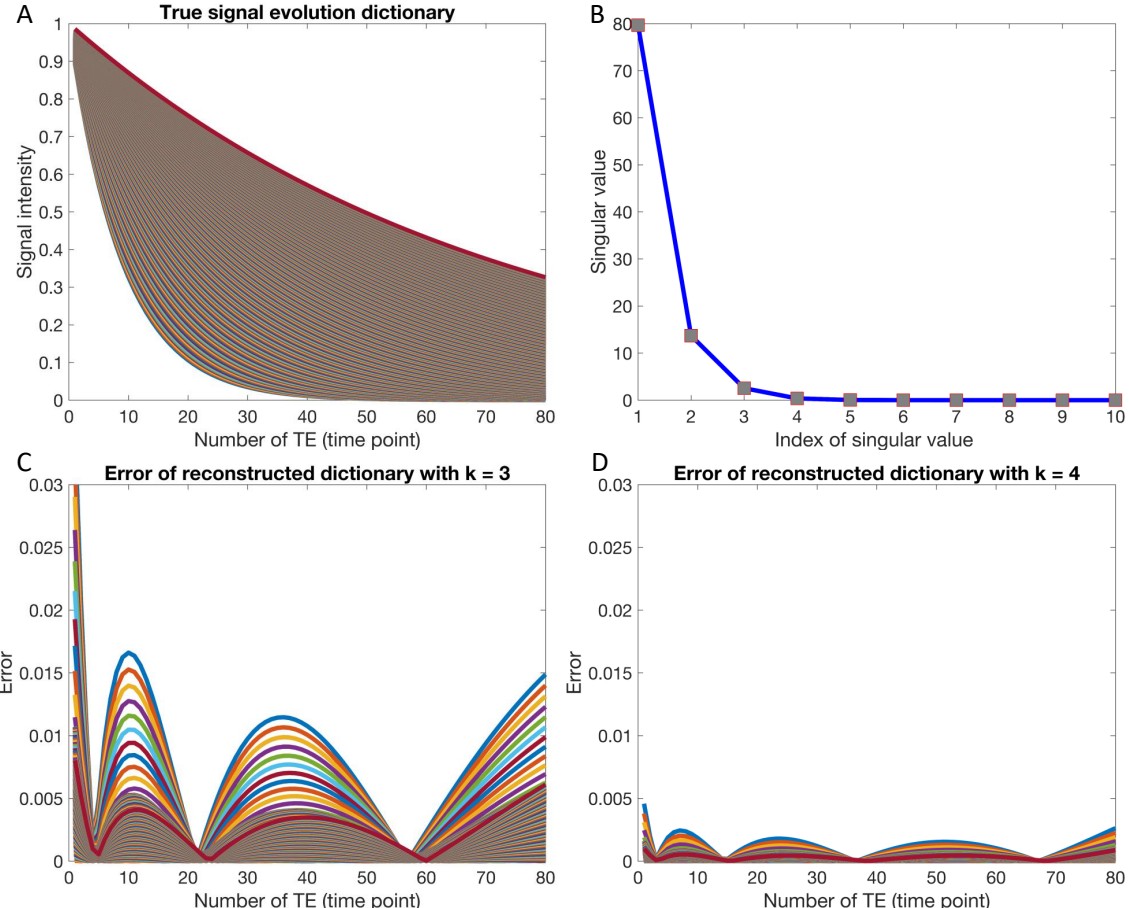

Figure 4: **(a)**: The Bloch signal evolution dictionary. **(b)**: The singular values of the Bloch signal evolution dictionary. **(c)**: The error of the reconstructed dictionary with $K = 3$. **(d)**: The error of the reconstructed dictionary with $K = 4$

Although more latent space bases could recover the signal dictionary better, with highly undersampled 2D FSE kspace, the conditioning of the reconstruction problem will get worse. As shown in fig. 5, $K = 3$ achieves better image quality and quantity, given the groundtruth sensitivity maps, for both T2 shuffling method and our proposed method. Both fig. 4 and fig. 5 show that $K = 3$ is adequate to recover the signal evolution and yield better conditioning for the reconstruction problem.

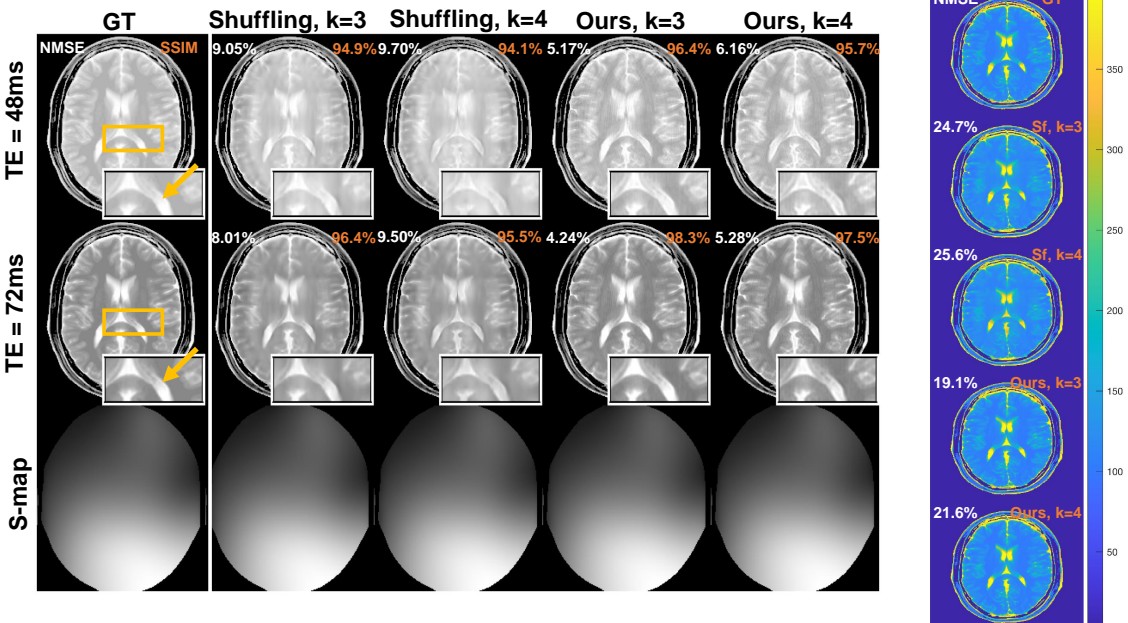

Figure 5: **The results of $K = 3$ and $K = 4$ for both T2 shuffling and our method**: $K = 3$ yields better image quality. For example, $K = 3$ recovers the zoomed in region much better. Meanwhile, our method still achieves better image quality and better performance on the NMSE and SSIM metrics. The error of the estimated T2 map is also smaller is $K = 3$ is used.

