# OpenReview forum: "Zero-Shot Self-Supervised Joint Temporal Image and Sensitivity Map Reconstruction via Linear Latent Space"
_MIDL.io/2023/Conference — MIDL 2023 Poster_

### Official Review · Reviewer_qquT · 2023-01-27

**Confidence:** 4
**Preliminary Rating:** 4
**Recommendation:** Poster

**Summary:**

The paper proposes an approach that utilizes a self supervised learning framework to learn a regularizer constrained on a
linear latent space which improves time-resolved FSE images reconstruction quality.

The proposed scheme is an extension of Aggarwal et al 2018, coupling DNN modeling with standard sparse modeling.


**Strengths:**


There is methodological novelty over prior work.

Practical results are interesting.

The method is evaluated over both simulated and real-world datasets.

The paper is well written. The empirical analysis is comprehensive.

**Weaknesses:**


The frameworks of Equations 3-4 and 5-6 (proposed) rely on heuristics, in the sense of the convergence point of the iterations isn't guaranteed. Each step of the roller is optimizing a completely new objective function. This is a major concern.



**Deanonymize Review:**

no

**Detailed Comments:**

please see the comments above.

**Paper Type:**

methodological development

**Questions To Address In The Rebuttal:**


The quality of the visualizations can be improved. As of now, it is absolutely unclear what qualitative improves the proposed makes.

The heuristical nature of the scheme remains a concern, and there is little promise that it can be resolved.

---

### Official Review · Reviewer_qcGF · 2023-02-02

**Confidence:** 4
**Preliminary Rating:** 3

**Summary:**

This paper describes a new approach for reconstructing time-resolved FSE MRI images.  The approach works by combining together previous self-supervised machine learning techniques with previous low-rank modeling techniques.  Simulations from retrospectively sampled data show a small improvement compared to existing methods.

**Strengths:**

The method is explained well and the paper is well-written and well-organized.  A small improvement has been demonstrated compared to existing techniques, and the validation study is properly scientific..

**Weaknesses:**

This paper seems like an incremental extension of existing techniques.  While there is a small performance improvement, there does not seem to be any conceptual innovation.  The paper also includes two different contributions (sensitivity map estimation and image reconstruction), which dilutes the message and makes it even harder to identify the innovation.

Using a dictionary with K=3 seems very restrictive for datasets with T=32 or T=80 echoes.  I have to imagine that this causes some non-negligible error?

The errors in the T2 map seem exceptionally large.  >20% error seems like it wouldn't be useful.  Was the shuffling implementation performed by the authors using code they built themselves, or was the original code used?  The shuffling results seem worse than I would have expected, and while the new method has a little improvement, the difference does not seem very consequential.


For the linear low-rank latent space model, it's good to also cite Liang's original work:

Liang ZP. Spatiotemporal imaging with partially separable functions. IEEE ISBI 2007, 988-991.

Haldar JP, Liang ZP. Low-rank approximations for dynamic imaging. IEEE ISBI 2011, 1052-1055.


**Deanonymize Review:**

no

**Paper Type:**

methodological development

**Questions To Address In The Rebuttal:**

Clarify the novel aspects of the paper.  If it's just numerical performance, then the contribution is less significant than if there was substantial conceptual novelty.

Consider ways to make the paper more focused.  Having two contributions (sensitivity maps and image reconstruction) makes the contribution harder to identify.

---

### Official Review · Reviewer_wNez · 2023-02-02

**Confidence:** 4
**Preliminary Rating:** 5
**Recommendation:** Oral, Poster

**Summary:**

The paper proposed a zero-shot self-supervised learning method to jointly reconstruct the temporal MR FSE images and the coil sensitivity maps via linear latent space using an iterative reconstruction technique. This methods have advantages of no training dataset and only using the self single data for network parameters update. It utilized the temporal linear latent model in T2 shuffling and self-supervised MR reconstruction learning method (SSDU). The results on simulated and retrospective in-vivo data showed that the proposed method was able to reconstruct highly-downsampled time-resolved FSE images.

**Strengths:**

The paper is well written. Compared with the widely proposed supervised learning methods for MR image reconstruction, the authors proposed a novel zero-shot self-supervised learning method for time-resolved FSE images reconstruction. Using the temporal linear latent space model proposed in T2 shuffling, the proposed network output the latent space rather than the MR images, which can better reconstruct the highly-downsampled time-resolved reconstruction problem. In addition, the authors splited the acquired k-space data to disjoint sub-kspaces for self-supervised learning and jointly reconstruct the MR images and the coil s-maps. The authors conducted experiments on simulated and in-vivo data to show that the proposed methods outperformed the conventional T2-shuffling and SSDU, and the way of simultaneously reconstruction of MR images and s-map is outperformed only performing the MR image reconstruction.

**Weaknesses:**

The authors divided the acquired kspace into two disjoint sub-kspaces for self-supervised learning reconstruction. It is unclear what the percentage of the spliting and how to split. It would be better to conduct some experiments to find the optimal way of split the kspace for self-supervised learning. Though the proposed method is zero-shot learning, it would be better to initialize the network with pretrained network using some large datasets (using supervised or self-supervised way), which might shorten the network converging time.

**Deanonymize Review:**

no

**Detailed Comments:**

Overall, the paper is clearly written and proposed a novel zero-shot self-supervised learning method to simultaneously learn the time-resolved MR images and coil s-maps. The authors conducted experiments which demonstrated the proposed methods outperformed the conventional T2-shuffling and self-supervised deep learning method SSDU. I suggested the authors to (1) clarify how they split the acquired kspace data for self-supervised learning and what the optimal way to split the acquired kspace data, (2) make the code and data open, (3) clarify how they choose the regularization parameters in the network loss.

**Paper Type:**

both

**Questions To Address In The Rebuttal:**

There are some questions needed to be addressed
1. how spliting acquired kspace into two disjoint sub-kspaces and why you choose it
2. what the size of  linear latent space Φ, and any effects on the size of latent space on the network performance
3. what the affects of regularization parameters (µ_I, µ_c, λ_c) in the network loss on the network performance
4. how you choose how many steps for each stage, and would better to plot curve showing the loss changes in the steps

---

### Meta-Review · Area_Chair_PikA · 2023-02-21

**Recommendation:** Accept (Poster)
**Confidence:** 5

**Metareview:**

The paper presents a novel method for reconstructing time-resolved MRI images from highly-undersampled FSE data. The learned self-regularization is an exciting idea. The paper address a relevant and significant problem in medical imaging acquisition. Results on simulated and in vivo datasets are presented.